# Alternative Microstructural Measures to Complement Diffusion Tensor Imaging in Migraine Studies with Standard MRI Acquisition

**DOI:** 10.3390/brainsci10100711

**Published:** 2020-10-06

**Authors:** Álvaro Planchuelo-Gómez, David García-Azorín, Ángel L. Guerrero, Rodrigo de Luis-García, Margarita Rodríguez, Santiago Aja-Fernández

**Affiliations:** 1Imaging Processing Laboratory, Universidad de Valladolid, 47011 Valladolid, Spain; aplanchu@lpi.tel.uva.es (Á.P.-G.); rodlui@tel.uva.es (R.d.L.-G.); sanaja@tel.uva.es (S.A.-F.); 2Headache Unit, Department of Neurology, Hospital Clínico Universitario de Valladolid, 47005 Valladolid, Spain; davilink@hotmail.com; 3Institute for Biomedical Research of Salamanca, 37007 Salamanca, Spain; 4Deparment of Medicine, Universidad de Valladolid, 47005 Valladolid, Spain; 5Deparment of Radiology, Hospital Clínico Universitario de Valladolid, 47005 Valladolid, Spain; margarita57@gmail.com

**Keywords:** migraine, chronic migraine, diffusion tensor imaging, magnetic resonance imaging (MRI), tract-based spatial statistics, diffusion magnetic resonance imaging

## Abstract

The white matter state in migraine has been investigated using diffusion tensor imaging (DTI) measures, but results using this technique are conflicting. To overcome DTI measures, we employed ensemble average diffusion propagator measures obtained with apparent measures using reduced acquisitions (AMURA). The AMURA measures were return-to-axis (RTAP), return-to-origin (RTOP) and return-to-plane probabilities (RTPP). Tract-based spatial statistics was used to compare fractional anisotropy, mean diffusivity, axial diffusivity and radial diffusivity from DTI, and RTAP, RTOP and RTPP, between healthy controls, episodic migraine and chronic migraine patients. Fifty healthy controls, 54 patients with episodic migraine and 56 with chronic migraine were assessed. Significant differences were found between both types of migraine, with lower axial diffusivity values in 38 white matter regions and higher RTOP values in the middle cerebellar peduncle in patients with a chronic migraine (*p* < 0.05 family-wise error corrected). Significantly lower RTPP values were found in episodic migraine patients compared to healthy controls in 24 white matter regions (*p* < 0.05 family-wise error corrected), finding no significant differences using DTI measures. The white matter microstructure is altered in a migraine, and in chronic compared to episodic migraine. AMURA can provide additional results with respect to DTI to uncover white matter alterations in migraine.

## 1. Introduction

Headache attacks in a migraine are characterized by episodes of unilateral pain of moderate to severe intensity, pulsating quality, aggravated by routine physical activity and accompanied by other symptoms, such as nausea and/or vomiting, photophobia and phonophobia, which last between 4 and 72 h [1]. Two main migraine types are currently distinguished: episodic migraine (EM) and chronic migraine (CM). The difference between both types is the frequency of headache days per month, which is 15 or more days in CM, and lower than 15 in EM, during at least three months [1].

To better understand migraine pathophysiology, diverse modalities of magnetic resonance imaging (MRI) have been employed. Among MRI modalities, those based on diffusion MRI (dMRI) give a particular insight on connectivity and white matter structure. Despite the advances of dMRI techniques, most of the migraine studies are based on the analysis of measures derived from diffusion tensor imaging (DTI). However, different DTI studies produce conflicting results. In most studies, the reported values of the fractional anisotropy (FA) were lower in migraine compared to controls in whole brain studies with tract-based spatial statistics (TBSS) [2,3,4,5], the most employed technique in dMRI migraine studies. Nonetheless, the opposite result, higher FA values in patients with a migraine, has also been found using the same assessment method [6]. In addition, only three dMRI studies with TBSS as assessment methods compared simultaneously patients with EM and CM, and controls. One study found significantly lower axial diffusivity (AD) values in CM compared to EM [7], another study found lower FA and higher mean diffusivity (MD) values in CM [8], and the other one found no significant differences [9].

To overcome DTI limitations, many different techniques have been proposed in the last decades, implying the acquisition of larger volumes of diffusion data (more gradient directions, more b-values) and, many times, longer processing times. Some examples of these techniques are multi-tensor models [10], Q-Ball imaging [10,11] or diffusion Kurtosis imaging (DKI) [12]. The trend over the last decade is the direct estimation of the ensemble average diffusion propagator (EAP), the probability density function of the motion of the water molecules inside each voxel [13,14]. The complete characterization of the EAP requires a large number of diffusion-weighted images with relative high b-values in a multishell acquisition. In clinical studies, the whole information provided by the EAP is translated into scalar values that can act as biomarkers. The most common measures are the return-to-origin (RTOP), return-to-plane (RTPP) and return-to-axis probabilities (RTAP) [15]. No dMRI studies with migraine patients have employed EAP-based measures.

Despite the advantages of the EAP-based measures, the calculation of these scalars usually requires long execution and acquisition times, together with very large b-values and a large number of diffusion gradients, not always available in commercial scanners and clinical routine. To solve these problems, a new methodology called “Apparent Measures Using Reduced Acquisitions” (AMURA) has been developed [16]. This tool allows the estimation of the EAP-related scalars without the explicit calculation of the EAP, using a lower number of samples, even with a single-shell acquisition scheme, assuming that the diffusion signal is independent from the radial direction. This methodology allows shorter MRI acquisition and very fast calculation of scalars. AMURA was initially designed for b-values of at least 2000 s/mm^2^, compatible with b-values employed in the high angular resolution diffusion imaging technique, which allows better modeling of white matter fiber architecture [11]. However, it could also be used for lower b-values, understanding that the effects measures by the scalars will be weaker.

Our objective was to assess whether EAP-based measures from the AMURA tool, calculated from a DTI compatible diffusion MRI acquisition (single-shell scheme and low b-value) typical in the clinical routine and in migraine diffusion MRI studies, was able to detect additional white matter changes between patients with migraine and controls with respect to DTI scalar measures.

## 2. Materials and Methods

### 2.1. Participants

A case-control study was carried out. Patients with a migraine were firstly screened and recruited from the headache outpatient unit at the Hospital Clínico Universitario de Valladolid (Valladolid, Spain). The participants from this study have been part of previous studies [7,17]. A total of 50 healthy controls (HCs), 54 patients with EM and 56 with CM were included in the sample. The inclusion criteria included diagnosis of EM or CM following the International Classification of Headache Disorders guidelines (third beta and third version) [1,18], stable situation of EM or CM in the preceding three months, agreement to participate in the study after signing the written informed consent and age from 18 to 60. The exclusion criteria included monthly frequency of a headache from 10 to 14 (exclusion of high frequency EM to avoid confusion with CM [19]), alternative craniofacial pain circumstances with a monthly frequency of 10 or higher, diagnosed major psychiatric disorders (in anamnesis or following the depression threshold from the Hospital Anxiety and Depression Scale [20]), additional neurological diseases or headache disorders, drug or substance abuse and pregnancy or childbearing. Every patient included in the sample was preventive naïve and fulfilled a headache diary the three months before inclusion. In some patients, a preventive treatment for migraines was prescribed at the visit. These patients started the prescribed prophylactic treatment after the MRI acquisition. Patients with EM from the sample suffered no tension-type headache. HC presented neither a present nor past history of migraines, nor major psychiatric or headache disorders, excluding infrequent tension-type headaches. No participants with brain abnormalities detected on T1-weighted MRI data by a radiologist were included in the sample. Patients were sampled following a non-probabilistic method by convenience sampling. Since the first patient (and first visit), all consecutive patients were informed and invited to participate, and enrolled if they agreed and signed the informed consent form. HC were balanced for age and sex by snowball and convenience sampling, following recruitment through advertisements in the University and hospital and colleagues.

Age and gender were gathered from every participant. The following characteristics were collected from every patient: duration of the migraine (years), monthly frequency of headache and migraine attacks (days), number of months from the onset of CM (if pertinent), presence of aura and intake of symptomatic medication for migraine (combination of analgesics and triptan). Acute medication overuse was considered if the intake monthly frequency was equal or higher than 10 according to the headache diary, following the International Classification of Headache Disorders guidelines (third beta and third version) [1,18].

The Hospital Clínico Universitario de Valladolid local Ethics Committee approved this study (PI: 14-197). Participants read and signed a written informed consent form before taking part in the study.

### 2.2. MRI Acquisition

All patients scanned suffered no migraine attacks in the previous 24 h. MRI acquisition was performed with a Philips Achieva 3 T MRI unit (Philips Healthcare, Best, The Netherlands), using a 32-channel head coil in the MRI facility at the University of Valladolid (Valladolid, Spain).

First, high-resolution 3D anatomical T1-weighted images were acquired using the following parameters: Turbo field echo (TFE) sequence, repetition time (TR) = 8.1 ms, echo time (TE) = 3.7 ms, flip angle = 8°, 256 × 256 matrix size, spatial resolution of 1 × 1 × 1 mm^3^ and 160 slices that cover the whole brain.

Then, diffusion-weighted data were obtained. The parameters employed in the acquisition were TR = 9000 ms, TE = 86 ms, flip angle = 90°, single-shell acquisition with 61 gradient directions and b-value = 1000 s/mm^2^, one baseline volume, 128 × 128 matrix size, spatial resolution of 2 × 2 × 2 mm^3^ and 66 slices that cover the whole brain.

Both T1 and diffusion-weighted data were collected between May 2014 and July 2018 in a unique MRI session, starting with the T1 scan. For a single subject, the time for both scans was approximately 18 minutes.

### 2.3. MRI Processing

#### 2.3.1. Diffusion MRI Preprocessing

The preprocessing steps were denoising, correction for eddy currents and motion and correction for B1 field inhomogeneity. The MRtrix software [21] was employed to carry out these steps, using the “dwidenoise”, “dwipreproc” and “dwibiascorrect” (-fast option) tools [22,23,24,25]. A whole brain mask for each subject was acquired with the “dwi2mask” tool [26].

#### 2.3.2. Calculation of the Diffusion Measures

Two groups of diffusion measures were employed. On the one hand, four measures from DTI were employed: FA, MD, AD and radial diffusivity (RD). On the other hand, three measures were computed with AMURA: RTOP, RTAP and RTPP.

The estimation of the diffusion tensor at each voxel, with the corresponding obtention of FA, AD and MD, was performed with the “dtifit” tool from the FSL software [27]. RD was manually obtained from the mean value of the second and the third eigenvalues from the diffusion tensor. FA manifests the directionality of water molecules displacement by diffusion, MD the average magnitude of water molecules diffusion, AD the water diffusion in the main direction of white matter fibers and RD the water diffusion in the perpendicular direction with respect to the main direction [28].

AMURA was employed to estimate the RTOP, RTPP and RTAP values. The tool can be downloaded with no restrictions in the following link: https://www.lpi.tel.uva.es/AMURA. We ran AMURA using MATLAB 2019b. The calculation with AMURA saves a great amount of time assuming that the diffusion signal is independent from the radial direction. Details about the mathematical models and the comparison against the whole EAP can be found elsewhere [16]. The RTOP has been pointed out as a better biomarker for cellularity and diffusion restrictions in comparison with the MD [29], the RTPP as a marker of diffusion restriction in the axial direction [14] and the RTAP as a marker of diffusion restriction in the radial direction [14]. A visual comparison of the three measures obtained with the AMURA tool, and the corresponding DTI measures, considering the commented inverse trends with the eigenvalues, is shown in Figure 1. It is worth noting that, assuming a simpler Gaussian diffusion propagator, the RTOP, RTPP and RTAP values can be calculated using the diffusion tensor: they are associated with the inverse values of the square root of the three eigenvalues, the first eigenvalue and the second and third eigenvalues, respectively. These equations are shown in Appendix A. Note that RTOP and RTAP depend on the inverse of the smaller eigenvalue. As a consequence, these two measures, when calculated using the diffusion tensor, are very sensitive to noise and outliers. In this study, we discarded this calculation, since it produces very high values in most of the areas of interest (high anisotropy) that makes any further analysis unfeasible. AMURA, on the other hand, produces robust values for the three considered measures.

#### 2.3.3. Tract-Based Spatial Statistics (TBSS)

TBSS was employed to compare the diffusion measures in white matter tracts between the three groups of interest [30]. The white matter tracts were identified according to the Johns Hopkins University ICBM-DTI-81 White Matter and the White Matter Tractography Atlas to cover the whole brain [31,32,33], considering a total of 48 and 20 regions of interest (ROIs), respectively. The first step of TBSS was the nonlinear registration of the participants’ FA images to a template of averaged FA images (FMRIB-58) in the Montreal Neurological Institute (MNI) space with the FNIRT tool [34]. After the registration, a mean FA skeleton image was created with an FA threshold value of 0.2 to distinguish white from gray matter. The individual FA images from the subjects were projected onto the mean FA skeleton, and the TBSS projection was repeated for the non-FA images, i.e., MD, AD, RD, RTOP, RTAP and RTPP. The minimum volume to consider significant results in a single region was 30 mm^3^, equal to the number of voxels in this study, which could be part of one or more clusters.

### 2.4. Statistical Analysis

For the analysis of the quantitative variables, normality and homogeneity of variance were assessed with the Kolmogorov–Smirnov and the Levene’s test for equality of variances, respectively. In the comparisons with the three groups, the parametric test employed was a one-way ANOVA, and the Kruskal–Wallis test if normality and homogeneity assumptions were rejected. For the comparisons between both migraine groups, two-tailed unpaired *t*-test and Mann–Whitney U test were used instead of ANOVA and Kruskal–Wallis tests, respectively. The comparison of gender between the three groups was performed with a chi-squared test, and comparisons of categorical features between both migraine groups were performed with the Fisher’s exact test.

Regarding the TBSS analysis, the “randomize” tool, a permutation-based inference tool by nonparametric statistics, using the threshold-free cluster enhancement (TFCE) option from FSL [35,36], was used to test the voxelwise differences between the three groups. The number of permutations for each comparison was 5000 to perform a robust inference, and *p* < 0.05, family-wise error corrected with the TFCE option, was the statistical threshold to consider significant results.

In a secondary analysis, the TBSS analysis was repeated including the time from onset of CM and the total duration of a migraine as covariates in separate comparisons. Both covariates were not included simultaneously in the analysis because of possible collinearity.

Moreover, the association between the duration of migraine (total duration or time from onset of CM) and DTI and AMURA measures was assessed. To analyze trends within each type of migraine and following the previous study with the same sample [7], we acquired the correlation values in patients with EM and CM independently. The inverse warp fields of the FA images to the MNI space transformation from the TBSS processing steps were obtained and used to obtain individual label maps based on the John Hopkins University ICBM-DTI-81 White Matter atlas. The mean value of each parameter in the diverse regions of the atlas and the Spearman’s rank correlation value were employed in the correlation analysis. The results were corrected for multiple comparisons with the Benjamini–Hochberg false discovery rate method [37]. A value of *p* < 0.05, adjusted for multiple comparisons, was considered statistically significant.

## 3. Results

Fifty HC, 54 patients with EM and 56 patients with CM were included in the sample. No significant differences in age or gender were found between the three groups. Patients with CM showed significantly higher duration of migraine, frequency of headache and migraine attacks and overuse of medication, and a lower presence of aura. The detailed characteristics from the three groups are shown in Table 1 (also shown in the previous studies with the same sample [7,17]).

### 3.1. TBSS

Using the DTI measures (FA, MD, AD and RD), the only comparison with significant differences was that between both groups of patients with a migraine. Patients with CM showed lower AD values than EM in 38 out of 48 regions from the ICBM-DTI-81 Atlas, and in 15 out of 20 regions from the White Matter Tractography Atlas. These results are depicted in Figure 2, and in Table 2 and Table 3.

With respect to the RTOP, RTAP and RTPP values, first, with the assumption of the Gaussian model, the dependence of RTOP and RTAP on the smallest eigenvalue produced a great number of outliers in the areas with high anisotropy that made the TBSS comparisons unfeasible. Hence, we only analyzed the three values using the AMURA tool. As in the DTI comparison mentioned previously, significant differences were found between both groups of migraine patients, with higher RTPP values in patients with CM, but significant results were found only in the middle cerebellar peduncle (significant volume = 370 mm^3^, minimum corrected *p* = 0.035, peak at x = −19, y = −45, z = −35 in the MNI space). The RTPP comparison between both migraine groups is depicted in Figure 3. Furthermore, significant differences between patients with EM and HC were found using the RTOP. Patients with EM showed lower RTOP values than HC in 24 of the assessed regions from the ICBM-DTI-81 White Matter Atlas, and in 11 regions from the White Matter Tractography Atlas. The RTOP results are shown in Figure 4 and Table 4 and Table 5. No significant results with the AMURA tool were found either between patients with CM and HC or using the RTAP measure.

A summary with the previous TBSS results can be found in Figure 5.

### 3.2. TBSS with Covariates

After the inclusion of the duration of migraine history as a covariate, lower AD values were found in CM compared to EM in the middle cerebellar peduncle (significant volume = 675 mm^3^, minimum corrected *p* = 0.028, peak at x = −20, y = −50, z = −32). The same result was obtained including the time from onset of CM as a covariate in 23 regions. Higher AD values were found in EM in comparison with HC including the duration of migraine history as a covariate in seven regions from the left hemisphere. Lower FA values were identified in CM compared to HC including the time from onset of CM as a covariate in 12 regions located in the right hemisphere and the corpus callosum. These results are shown in Table A1, Table A2 and Table A3 (extracted from [7]).

Regarding the EAP-based measures, including the duration of the migraine as a covariate, a very similar result was obtained with the RTPP with respect to the analysis with no additional covariates, finding higher RTPP values in CM compared to EM in the middle cerebellar peduncle (significant volume = 351 mm^3^, minimum corrected *p* = 0.039, peak at x = −20, y = −50, z = −32 in the MNI space). Additionally, lower RTPP values were found in patients with EM in comparison with HC in eight regions from the ICBM-DTI-81 White Matter Atlas, with seven regions from the left hemisphere. Including the time from onset of CM as a covariate, lower RTAP values were found in CM compared to HC in four regions from the right hemisphere included in the ICBM-DTI-81 White Matter Atlas. These results are shown in Table A4 and Table A5 and Figure A1 and Figure A2.

Taking into account the significant differences of the presence of an aura and a medication overuse headache between patients with CM and EM, we additionally included both variables in the analysis with multiple covariates, together with the duration of the migraine history.

Including only the presence of an aura as a covariate, with respect to the results with no covariates, AD values were lower in CM than in EM in eight regions, the RTPP was higher in CM than in EM in the middle cerebellar peduncle and no significant RTOP differences between EM and HC were obtained. No additional significant results were identified.

Regarding significant differences between CM and EM in the multivariate model, AD (middle cerebellar peduncle), MD (34 regions) and RD (39 regions) values were lower in CM. The opposite statistically significant trend, i.e., higher values in CM compared to EM, was obtained for RTPP (15 regions), RTOP (42 regions) and RTAP (38 regions). These results are shown in Table A6, Table A7, Table A8, Table A9, Table A10 and Table A11 and Figure A3, Figure A4 and Figure A5. Similar results, with the same significant statistical comparisons but a different number of regions with significant differences, were obtained including only a medication overuse headache as a covariate. In addition to the previous results, including only medication overuse headache as a covariate, FA values were higher in CM compared to EM in the body (60 voxels, minimum adjusted *p* = 0.047) and splenium (136 voxels, minimum adjusted *p* = 0.040) of the corpus callosum, the left superior corona radiata (68 voxels, minimum adjusted *p* = 0.047) and the left tapetum (35 voxels, minimum adjusted *p* = 0.040).

Statistically significant differences employing the model with the three covariates were obtained between EM and HC. On the one hand, increased AD (eight regions) values were found in EM with respect to HC. On the other hand, reduced RTPP (five regions) values were identified in EM compared to HC. These results are shown in Table A12 and Table A13 and Figure A6.

In the comparison between CM and HC, significantly higher FA (10 regions) values were found in CM, and significantly lower RD (14 regions) values were identified in CM. These results are shown in Table A14 and Table A15 and Figure A7.

### 3.3. Correlation Analysis

Statistically significant positive correlation was identified between time from onset of CM and mean FA in the left (ρ = 0.439, unadjusted *p* < 0.001) and right (ρ = 0.420, unadjusted *p* = 0.001) external capsule after the correction for multiple comparisons. Statistically significant negative correlation was found between time from onset of CM and mean RD in the left (ρ = −0.439, unadjusted *p* < 0.001) and right (ρ = −0.427, unadjusted *p* = 0.001) external capsule. The significant correlation results are shown in Figure 6 (extracted from [7]).

No additional statistically significant correlations were found for mean AD or MD, the three EAP-based measures, or the total duration of the migraine in patients with EM or CM.

## 4. Discussion

Two main results were obtained with the AMURA tool in comparison with DTI scalar measures, using a dMRI acquisition protocol typical in the clinical routine and migraine studies. On the one hand, with AMURA, significant differences between patients with CM and EM were obtained with the RTPP, a result provided by AD and DTI. However, the number of regions with significant differences was lower with the RTPP. This lower sensitivity of AMURA was motivated by the fact that RTPP was designed to work with higher b-values. On the other hand, additional significant differences between patients with EM and HC were obtained only with AMURA based on the RTOP results. These results suggest that the AMURA tool may be a great complement to DTI in dMRI studies with migraine patients. We expect that the differences would grow if higher b-values were used.

Regarding the calculation of the EAP-based measures, when using the eigenvalues of the diffusion tensor, we were unable to perform statistical comparisons between the groups with TBSS because the values of RTOP and RTAP in some voxels were extremely high. The very low values of the two smallest eigenvalues in specific voxels explain the inability to work with real values of RTOP and RTAP and DTI. Therefore, the DTI approach is excessively simple to work with EAP-based measures, and tools such as AMURA are needed, especially in those studies with time restrictions and one single b-value.

White matter differences obtained with AMURA are in line with the TBSS results reported with the same sample in a previous study [7]. Lower AD and higher RTPP values in patients with CM compared to EM suggest axonal damage or loss, as previously suggested by Yu et al. [38], or short-term demyelination [39,40,41]. Anyway, results from DTI scalar measures must be interpreted with caution, considering that the relationship between DTI parameters and microstructural alterations are not completely clear.

The significant differences between patients with CM and EM obtained with the RTPP were a subsample of the results obtained with the AD. It must be considered that the diffusion MRI acquisition protocol was suboptimal for the use of the AMURA tool because of the low b-value and single-shell scheme, and, possibly for that reason, we were unable to identify the same or a similar number of significant differences between the migraine groups. The three measures obtained with the AMURA tool are designed to measure effects related to b-values over 2000 s/mm^2^ (our b-value was 1000 s/mm^2^), and RTPP is particularly sensitive to the b-value. Therefore, the RTPP result reflects the potential of AMURA to identify white matter structural changes in migraines using an acquisition protocol more appropriate for the tool.

RTPP differences between patients with CM and EM were found in the middle cerebellar peduncle. The middle cerebellar peduncle connects the cerebellum to the pons. Smaller cerebellar volume has been identified in patients with CM compared to HC [42]. Moreover, it has been suggested that the cerebellum suffers a neuropathological change in a migraine related to spreading depression [43]. The dorsolateral pons has been shown to be activated during a migraine and potentially involved in other mechanisms such as transmission of nociceptive signals to the hypothalamus, amygdala and basal forebrain [44]. Another study by Chong et al. has reported significant deformation of the pons in patients with a migraine [45]. Our results may suggest that the connectivity between the cerebellum and the pons is altered in CM compared to EM, possibly in association with structural changes of these regions linked to the migraine experience.

With respect to the comparison between patients with migraine (EM in this study) and controls, we only obtained significant differences using AMURA. This result reflects that AMURA may be helpful to determine additional microstructural changes between patients with migraine and controls with respect to the single use of DTI.

The lower RTOP values found in patients with EM compared to HC are in line with the most reported result with the MD in previous studies (higher MD in migraines) [2,5,46,47,48,49,50,51], although the opposite result with the MD (lower MD in migraines) has also been reported [3,6,38,52]. It must be noted that most patients included in previous migraine studies were patients with EM. In our study, only patients with low frequency EM (less than 10 headache days per month) were included to assure that there were no patients close to the CM frequency threshold.

Some of the white matter regions with higher MD reported previously were identified as regions with significant differences between EM and HC in our study. These regions are the anterior thalamic radiation, the inferior longitudinal fasciculus, the corticospinal tract, the corpus callosum (genu and forceps minor) and the inferior cerebellar peduncle [5,47,49,50]. Higher MD values have been associated with edema and Wallerian degeneration [39]. The changes in the anterior thalamic radiation may be associated with structural connections with the thalamus, which has been reported as a key region in migraine pathophysiology. In migraines, the thalamus has been associated with allodynia, photophobia and photoallodynia [53,54] and it has been suggested to have a role in the abnormal functional connectivity in diverse brain networks in the interictal state [55]. The corticospinal tract has been described as an important white matter region in relationship with nociceptive perception [56]. The inferior longitudinal fasciculus connects regions from the temporal and occipital cortex. One of the regions that takes part in the networks connected by the inferior longitudinal fasciculus is the temporal pole, which has shown hyperactivation in functional MRI studies with migraine patients, including connections with the thalamus and the insula [57,58], and loss of gray matter volume and altered cortical thickness [59,60]. Our results and the previous findings suggest that changes in the structural white matter connectivity may be associated with functional connectivity and gray matter alterations in migraine patients.

Regarding previous DTI results in migraines, the reduction of FA in patients with a migraine compared to controls is the most frequent result in diffusion MRI studies [2,3,4,5,38,46,49,61,62,63,64,65], but the opposite result has also been reported in pediatric patients by Messina et al. [6] and in the thalamus by Coppola et al. [66]. In this study, we obtained both higher and lower FA values in patients with CM compared to controls, associated with medication overuse headache and duration of the migraine, respectively. Most regions related to both results were different, and the related clusters were not extremely close. A possible reason of these apparently contradictory results would be the coexistence of debilitated and enhanced structural networks in the migraine, as it has been previously suggested [17], which may partially explain the apparently conflicting results in the literature. According to our results, these networks would be related to different pathophysiological mechanisms associated with medication overuse and longitudinal effects. In patients with EM and a medication overuse headache, lower FA values have been reported compared to controls [5]. Additional significant differences were obtained between CM, EM and HC including the duration of the migraine and medication overuse headache, but no changes were specifically related to the presence of an aura, in contrast to previous studies [67]. The lack of significant results associated with an aura in our study may be caused by the relative low number of patients with an aura. Future studies should specifically analyze differences between patients with and without medication overuse headache in patients with CM, and longitudinal studies should be performed to assess the longitudinal effects of CM in white matter.

With respect to the discordance with previous studies, there could be additional reasons. One of the reasons would be methodological. For example, in the study by Rocca et al. (2003), the analyzing method consisted of the study of histogram peaks [52], which is considerably different from methods carried out in the most recent years. Another reason would be associated with different sample characteristics. In the study by Messina et al., the sample was composed of pediatric patients [6], and the changes in the brain might be different compared to the alterations in adults. The sample by Yu et al. [3] contained patients with depression, which might have influenced the results, considering that our sample included no patients with anxiety or depression.

Regarding the alterations found in this study, it should be elucidated whether the identified changes were migraine-specific. In a previous study including 277 headache free subjects and 246 patients with headaches, including 69 patients with migraines and 76 with tension-type headaches, Kattem Husøy et al. found no significant TBSS differences between migraines and tension-type headaches [49]. In the same study, the authors identified significantly higher AD values in patients with migraines and tension-type headaches compared to headache free subjects, with a higher number of voxels with significant differences in the migraine. Furthermore, patients with any headaches and a new onset headache presented widespread higher AD, MD and RD values compared to controls in the aforementioned study. These results suggest alterations in patients with a headache compared to controls, but with no clear migraine biomarkers in contrast to other headache disorders. The specific migraine microstructural brain changes in comparison with other headache and pain disorders should be analyzed in future studies, in order to uncover the particular pathophysiological characteristics of a migraine. Another aspect that needs to be studied is whether the identified changes are the cause or consequence of the migraine.

There were some limitations worth mentioning in this study. Due to time restrictions related to the MRI acquisition protocol, it was not possible to collect T2 or T2-FLAIR MRI data to assess the presence of white matter hyperintensities. Pain in patients with EM and an adverse prognosis have been related to white matter hyperintensities in migraines [68,69]. Considering the relative high risk of detection of white matter hyperintensities on MRI in migraines and their negative impact, the presence of these lesions might have some influence on the results, although the associated pathophysiology and long-term effects are unclear [70]. The baseline volume from the diffusion MRI acquisition is similar to a T2-weighted image, although its quality is low for the assessment of white matter hyperintensities. In relationship with the relative low acquisition, we performed no multishell acquisition with moderate-to-high b-values, a better scenario for the EAP-based measures, using the AMURA tool or the whole EAP. The absence of diffusion data with higher b-values prevented us from exploiting the full potential of the AMURA tool, although it allowed us to explore its discriminating power in conditions frequent in other migraine clinical MRI studies. The lack of control of the time to the next migraine attack might have influenced our results because some patients were in prodromal instead of interictal state when the MRI data were acquired, and alterations of brain physiology and function have been observed during the prodromal stage [71,72]. Another fact that might have influenced our results is the medication overuse, which was present in most patients with CM (75%), although we corrected the results considering this variable. The presence of anxiety and/or depression, which are frequent in migraines, might influence brain connectivity, as pointed by previous studies. Indeed, a smaller brain volume has been associated with depression in migraine, and migraine with depression may represent a different clinical phenotype with a specific long-term evolution [73]. In our sample, however, there were no patients with anxiety and/or depression. Although this fact prevented us from this possible relationship, the absence of patients with depression or anxiety also avoided the possible bias in the results that could have been caused by the inclusion of this distinct phenotype. The diagnosis of infrequent tension-type headache was not performed using the headache diary, but only the history, which may be inaccurate to determine more than a unique associated tension-type headache day per month, or annual frequency higher than 12 days.

## 5. Conclusions

EAP-based measures obtained with the AMURA tool could detect white matter changes in patients with migraine to complement the results obtained with DTI scalar measures using diffusion MRI protocols with a single-shell acquisition low b-value, which are typical in the clinical routine and migraine clinical studies. Our results support structural connectivity changes between patients with EM and CM, and changes in the brain white matter related to migraine. Future studies should employ diffusion MRI multishell acquisitions with moderate-to-high b-values, when possible, in order to exploit the full potential of AMURA and identify white matter changes in patients with migraine. Other research lines may include the interaction between changes in white matter connectivity, gray matter structure and functional connectivity.

## Figures and Tables

**Figure 1 brainsci-10-00711-f001:**
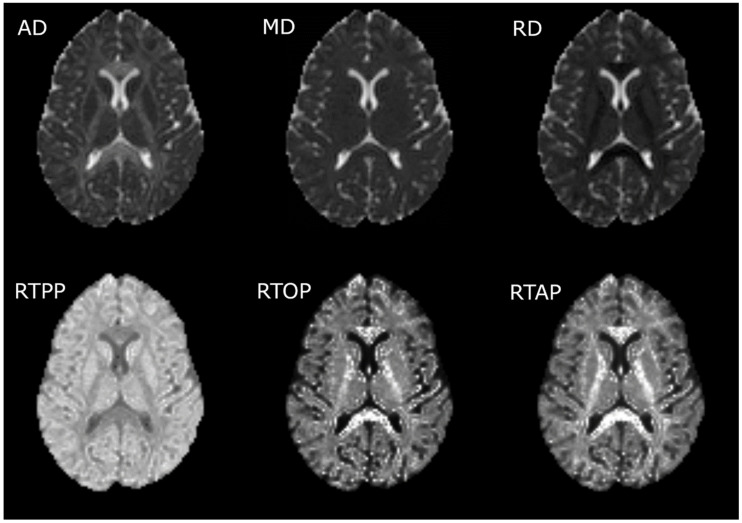
Visual comparison of diffusion tensor imaging (DTI) and measures from apparent measures using reduced acquisitions (AMURA). The first row contains the DTI measures, and the second row the AMURA metrics. It is worth noting that the brightest regions for the DTI measures correspond to the darkest regions for the AMURA metrics, and vice versa.

**Figure 2 brainsci-10-00711-f002:**
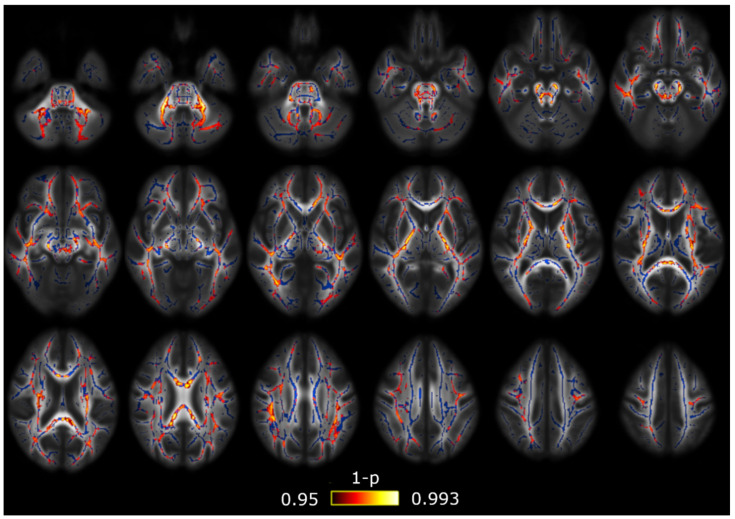
Axial diffusivity (AD) alterations in patients with CM in comparison with patients with EM. Widespread significant lower AD values were found in CM. The white matter skeleton is shown in blue and voxels with significant differences in red-yellow. The color bar shows the 1-p values (family-wise error corrected).

**Figure 3 brainsci-10-00711-f003:**
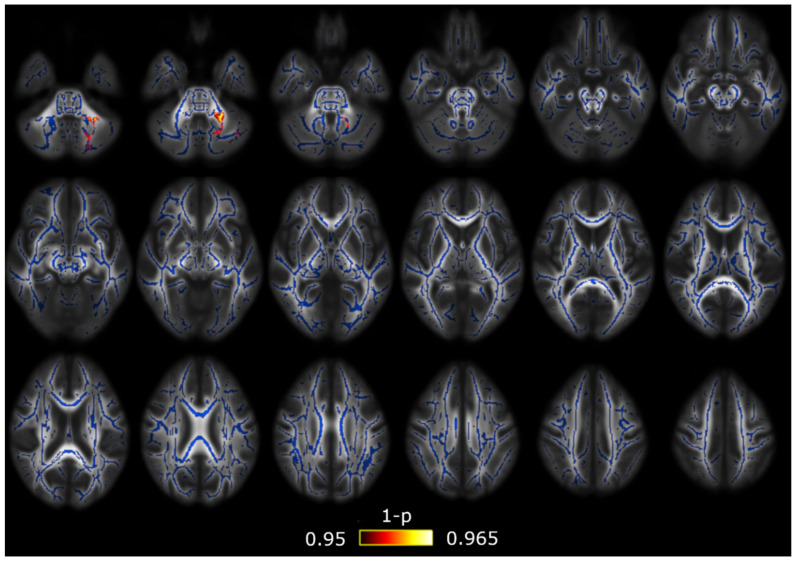
Return-to-plane (RTPP) alterations in patients with CM in comparison with patients with EM. Significant higher RTPP values in CM were found only in the middle cerebellar peduncle. The white matter skeleton is shown in blue and voxels with significant differences in red-yellow. The color bar shows the 1-p values (family-wise error corrected).

**Figure 4 brainsci-10-00711-f004:**
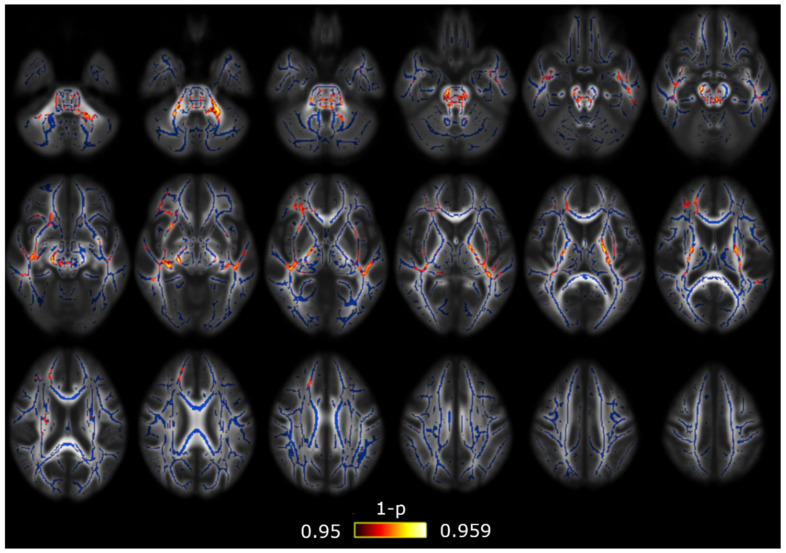
Return-to-origin (RTOP) alterations in patients with EM in comparison with HC. Lower RTOP values were found in EM. The white matter skeleton is shown in blue and voxels with significant differences in red-yellow. The color bar shows the 1-p values (family-wise error corrected).

**Figure 5 brainsci-10-00711-f005:**
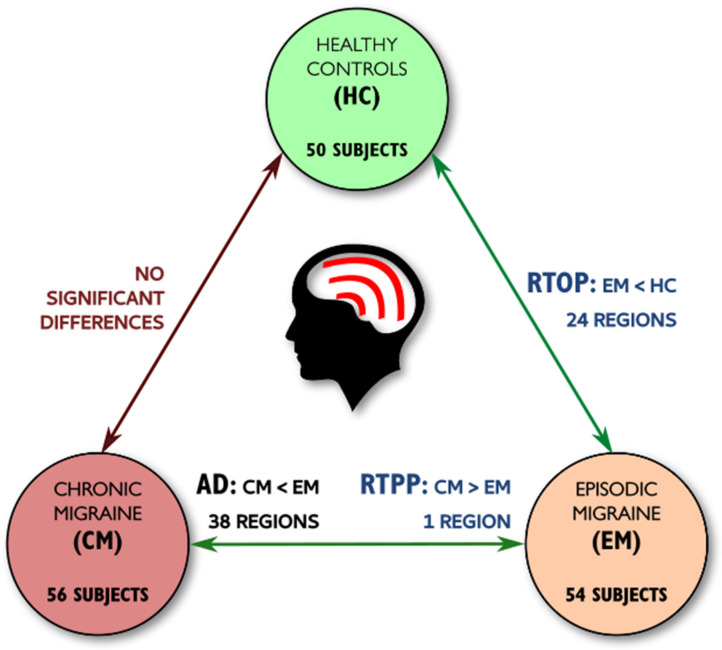
Summary of the main tract-based spatial statistics (TBSS) results with DTI and AMURA measures. Significant differences between any migraine group and controls were found only with the RTOP from AMURA, identifying white matter alterations non-measurable with DTI. Differences between CM and EM were found with RTPP from AMURA, and AD from DTI. No differences were found between CM and controls.

**Figure 6 brainsci-10-00711-f006:**
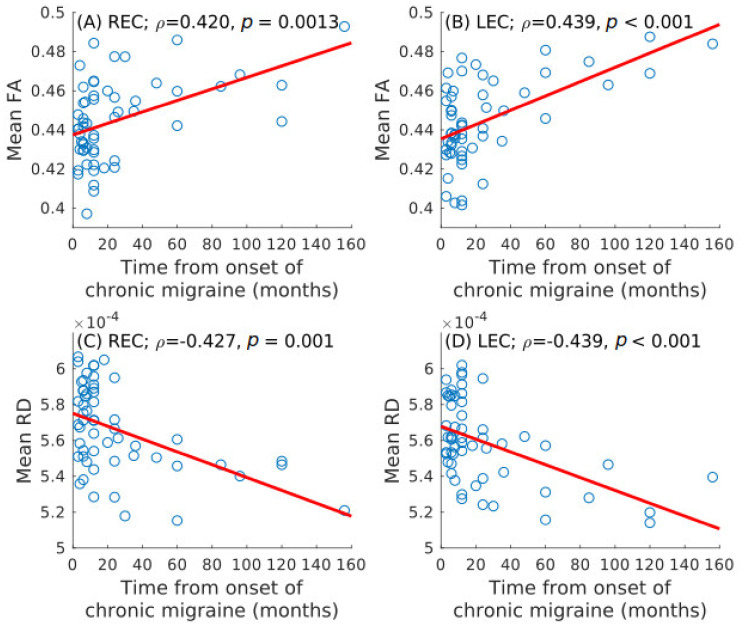
Spearman correlation values between the time from onset of CM and diffusion measures. Statistically significant positive association with the mean fractional anisotropy (FA) in the bilateral external capsule is shown in (**A**) and (**B**). Statistically significant negative association with the mean radial diffusivity (RD) is shown in (**C**) and (**D**). LEC = left external capsule; REC = right external capsule.

**Table 1 brainsci-10-00711-t001:** Clinical and demographic characteristics of healthy controls (HCs), and patients with an episodic migraine (EM) and chronic migraine (CM).

	HC (*n* = 50)	EM (*n* = 54)	CM (*n* = 56)	Statistical Test
Gender, male/female	11/39 (22/78%)	9/45 (17/83%)	6/50 (11/89%)	χ^2^_(2, N = 160)_ = 2.48, *p* = 0.29 ^†^
Age (years)	36.1 ± 13.2	37.1 ± 8.2	38.1 ± 8.7	χ^2^ (2) = 2.85, *p* = 0.24 ^‡^
Duration of the migraine history (years)		14.1 ± 11.1	19.6 ± 10.4	t_(108)_ = −2.7, *p* = 0.008 ^§^
Time from onset of CM (months)			24.5 ± 32.9	
Headache frequency (days/month)		3.6 ± 1.9	23.3 ± 6.3	U = 44.0, *p* < 0.001 ^⁋^
Migraine frequency (days/month)		3.6 ± 1.9	13.9 ± 6.9	U = 108.5, *p* < 0.001 ^⁋^
Overusing medication		0 (0%)	42 (75%)	*p* < 0.001 ^⁑^
Aura		9 (17%)	1 (2%)	*p* = 0.007 ^⁑^

^†^ Chi-square test. ^‡^Kruskal–Wallis test. ^§^Two-tailed, unpaired Student’s *t*-test. ^⁋^Mann–Whitney U test. ^⁑^Fisher’s exact test. Data are expressed as means ± SD.

**Table 2 brainsci-10-00711-t002:** White matter regions from the ICBM-DTI-81 White Matter Atlas for which significant decreased AD values were found in CM compared to EM.

White Matter Region	Minimum *p*-Value (FWE-Corrected)	Volume (mm^3^)	MNI Peak Coordinate (mm), (x,y,z)
Middle cerebellar peduncle	0.007	2206	(−20,−50,−32)
Superior cerebellar peduncle R/L	0.020/0.020	142/126	(5,−28,−19)/(−4,−28,−19)
Inferior cerebellar peduncle R/L	0.019/0.009	75/89	(12,−43,−35)/(−13,−45,−31)
Superior longitudinal fasciculus R/L	0.021/0.021	971/874	(33,−4,20)/(−36,−49,15)
Genu of corpus callosum	0.019	455	(10,−28,1)
Body of corpus callosum	0.032	842	(−4,30,23)
Splenium of corpus callosum	0.025	873	(22,−50,25)
Anterior corona radiata R/L	0.024/0.018	556/805	(18,21,−11)/(−18,38,−1)
Superior corona radiata R/L	0.020/0.022	666/396	(28,−16,21)/(−27,−11,20)
Posterior corona radiata R/L	0.022/0.022	201/214	(25,−24,24)/(−30,−52,22)
External capsule R/L	0.020/0.018	459/695	(30,−10,14)/(−22,16,−12)
Posterior limb of internal capsule R/L	0.020/0.022	569/536	(26,−17,13)/(−27,−17,17)
Retrolenticular part of internal capsule R/L	0.023/0.023	457/344	(31,−34,15)/(−25,−22,3)
Anterior limb of internal capsule R/L	0.022/0.020	216/290	(15,−1,7)/(−20,18,3)
Sagittal stratum R/L	0.022/0.022	471/359	(37,−49,−4)/(−41,−18,−13)
Posterior thalamic radiation R/L	0.022/0.022	353/279	(37,−50,−2)/(−35,−52,13)
Cerebral peduncle R/L	0.020/0.022	234/265	(11,−23,−21)/(−9,−19,−20)
Corticospinal tract R/L	0.019/0.023	106/165	(10,−27,−26)/(−7,−18,−22)
Medial lemniscus R/L	0.020/0.015	82/103	(8,−39,−40)/(−7,−37,−40)
Pontine crossing tract	0.018	82	(8,−31,−27)
Fornix (cres) R/L	0.024/0.024	74/45	(35,−12,−14)/(−34,−15,−13)
Cingulum (hippocampus) L	0.036	56	(−17,−42,−2)

FWE = Family-wise error; L = left; R = right. The column Volume represents the volume from the atlas region with significant results. No regions with a volume equal or lower than 30 mm^3^ were included in this Table.

**Table 3 brainsci-10-00711-t003:** White matter regions where significant decreased AD values were found in CM compared to EM using the White Matter Tractography Atlas.

White Matter Region	Minimum *p*-Value (FWE-Corrected)	Volume (mm^3^)	MNI Peak Coordinate (mm), (x,y,z)
Anterior thalamic radiation L/R	0.020/0.021	316/232	(−21,18,3)/(9,−29,−14)
Corticospinal tract L/R	0.022/0.018	627/601	(−24,−20,9)/(10,−28,−26)
Cingulum (hippocampus) L	0.036	37	(−17,−43,−2)
Forceps major	0.024	375	(−18,−85,8)
Forceps minor	0.018	1601	(−17,39,−2)
Inferior fronto-occipital fasciculus L/R	0.018/0.022	994/973	(−23,27,3)/(37,−49,−4)
Inferior longitudinal fasciculus L/R	0.022/0.022	418/507	(−35,−52,12)/(44,−33,−12)
Superior longitudinal fasciculus L/R	0.021/0.021	1023/828	(−36,−50,14)/(31,−6,17)
Superior longitudinal fasciculus (temporal part) R	0.022	62	(49,−33,−11)
Uncinate fasciculus L	0.018	83	(−18,21,−9)

FWE = Family-wise error; L = left; R = right. The column volume represents the volume from the atlas region with significant results. No regions with a volume equal or lower than 30 mm^3^ were included in this table.

**Table 4 brainsci-10-00711-t004:** White matter regions from the ICBM-DTI-81 White Matter Atlas for which significant decreased RTOP values were found in EM compared to HC.

White Matter Region	Minimum *p*-Value (FWE-Corrected)	Volume (mm^3^)	MNI Peak Coordinate (mm), (x,y,z)
Middle cerebellar peduncle	0.042	908	(13,−30,−26)
Superior cerebellar peduncle R/L	0.042/0.042	65/65	(7,−32,−19)/(−5,−31,−18)
Inferior cerebellar peduncle L	0.045	96	(−13,−45,−31)
Genu of corpus callosum	0.047	44	(18,31,15)
Anterior corona radiata R	0.044	446	(26,35,−1)
Superior corona radiata R	0.048	84	(23,−12,19)
External capsule R/L	0.042/0.046	415/392	(33,−19,−2)/(−33,−13,1)
Posterior limb of internal capsule R/L	0.042/0.046	285/420	(20,−20,−4)/(−22,−8,14)
Retrolenticular part of internal capsule R/L	0.041/0.044	244/314	(37,−26,−2)/(−37,−34,2)
Anterior limb of internal capsule L	0.046	43	(−17,−2,12)
Sagittal stratum R/L	0.041/0.044	243/97	(39,−29,−5)/(−40,−29,−6)
Cerebral peduncle R	0.042	254	(12,−25,−21)
Corticospinal tract R/L	0.042/0.042	156/135	(11,−25,−22)/(−7,−25,−26)
Medial lemniscus R/L	0.045/0.046	83/87	(9,−32,−25)/(−4,−37,−30)
Pontine crossing tract	0.042	163	(−4,−30,−28)
Fornix (cres) R	0.041	148	(32,−22,−6)
Uncinate fasciculus R	0.043	32	(35,−4,−14)

FWE = Family-wise error; L = left; R = right. The column volume represents the volume from the atlas region with significant results. No regions with a volume equal or lower than 30 mm^3^ were included in this table.

**Table 5 brainsci-10-00711-t005:** White matter regions where significant decreased RTOP values were found in EM compared to HC using the White Matter Tractography Atlas.

White Matter Region	Minimum *p*-Value (FWE-Corrected)	Volume (mm^3^)	MNI Peak Coordinate (mm), (x,y,z)
Anterior thalamic radiation L/R	0.046/0.042	34/68	(−7,−36,−27)/(9,−29,−14)
Corticospinal tract L/R	0.042/0.042	284/374	(−7,−25,−26)/(11,−25,−22)
Forceps minor	0.046	342	(19,38,16)
Inferior fronto-occipital fasciculus L/R	0.044/0.044	240/676	(−39,−30,−4)/(37,−27,−3)
Inferior longitudinal fasciculus L/R	0.043/0.042	309/167	(−41,−28,−4)/(42,−14,−14)
Uncinate fasciculus L/R	0.048/0.044	62/71	(−35,−2,−20)/(34,1,−16)

FWE = Family-wise error; L = left; R = right. The column volume represents the volume from the atlas region with significant results. No regions with a volume equal or lower than 30 mm^3^ were included in this Table.

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
