# Peer review of "Alternative Microstructural Measures to Complement Diffusion Tensor Imaging in Migraine Studies with Standard MRI Acquisition"

_brainsci, 2020, doi:10.3390/brainsci10100711_

Round 1
Reviewer 1 Report
The present study investigates a particular method of assessing white matter changes (AMURA). The authors report results which are supportive of structural connectivity changes in migraine confirmed with this method of analysis.
The study is ambitious in scope and well executed. The manuscript is well written and the conclusions sound. I have only minor comments.
Good introduction.
Snowball sampling was used for controls. How were the first participants recruited?
Was medication overuse assessed based on the diary or anamnesis?
It is stated that the patients were preventive naïve. This is impressive considering the duration of the disease in some. How was it assessed?
How would white matter hyperintensities, which were not assessed, influence results. Please elaborate.
Very few patients had aura. Why not exclude them?
Was any standard, structural/clinical assessment of scans performed? Ie. were there any radiological findings?
The authors state that the sample did not contain patients with depression or anxiety (pg 12 ln 351) yet on pg 13 ln 366 it is stated that the presence of this was not assessed. It is not completely clear what is meant.
Author Response
Reviewer #1: The present study investigates a particular method of assessing white matter changes (AMURA). The authors report results which are supportive of structural connectivity changes in migraine confirmed with this method of analysis.
The study is ambitious in scope and well executed. The manuscript is well written and the conclusions sound. I have only minor comments.
Good introduction.
We thank the reviewer the valuable comments. We have clarified some aspects that were unclear thanks to the comments.
- Snowball sampling was used for controls. How were the first participants recruited?
R1.1. Thank you for the question. The first controls were recruited through hospital and University colleagues and advertisements in these facilities by convenience sampling. We have clarified this point. With respect to the patients, since the first patient – first visit, we used a probabilistic sampling method and all consecutive patients were informed and invited to participate, and enrolled if they agreed and signed the informed consent form.
[Page 3, lines 107-112] “Patients were sampled following a non-probabilistic method by convenience sampling. Since the first patient (and first visit), all consecutive patients were informed and invited to participate, and enrolled if they agreed and signed the informed consent form. HC were balanced for age and sex by snowball and convenience sampling, following a recruitment through advertisements in the University and hospital and colleagues.”
- Was medication overuse assessed based on the diary or anamnesis?
R1.2. The medication overuse assessment was based on the diary. Medication overuse was considered in case of intake of analgesics and/or triptans during 10 or more days per month, following the guidelines from the International Classification of Headache Disorders (ICHD-3 beta and ICHD-3 versions, depending on the most recent available version). We have remarked the use of the diary for the medication overuse analysis.
[Page 3, lines 116-119] “Acute medication overuse was considered if the intake monthly frequency was equal or higher than 10 according to the headache diary, following the International Classification of Headache Disorders guidelines (third beta and third version) [1,18].”
- It is stated that the patients were preventive naïve. This is impressive considering the duration of the disease in some. How was it assessed?
R1.3. We agree with the reviewer. It may be surprising that some patients suffering from migraine for many years have never received a preventive treatment. The reason why that was possible is that our headache unit receives patients both from secondary-tertiary care and directly from primary care. The waiting list is around one month, and, in some cases, general practitioners refer patients without starting a prophylactic. Despite this is not the ideal situation for patients, it allows us to study this particular phase of the migraine disease in treatment naïve patients. In our setting, around 60% of patients with episodic migraine and 40% of patients with chronic migraine are prophylactic naïve. The therapies previously used were assessed by anamnesis and general practitioner recordings. Regarding the patients with chronic migraine, most patients suffered from chronic migraine for less than two years (please see the new Figure 6 to observe the individual time from onset of chronic migraine in our sample). Therefore, a preventive treatment was prescribed for most of these patients, but they started it after the MRI acquisition. We have added the most relevant information in the manuscript:
[Page 3, lines 101-104] “Every patient included in the sample was preventive naïve and fulfilled a headache diary the three months before inclusion. In some patients, a preventive treatment for migraine was prescribed at the visit. These patients started the prescribed prophylactic treatment after the MRI acquisition.”
- How would white matter hyperintensities, which were not assessed, influence results. Please elaborate.
R1.4. White Matter Hyperintensities (WMH) have been associated with pain in patients with episodic migraine and an unfavorable prognosis, although the pathophysiology of WMH is still unclear. Considering the relative high prevalence of WMH in migraine and the possible effects of these lesions, WMH might have influenced the results. Furthermore, the demographic profile of the sample suggests that it is uncommon the presence of microvascular lesions or other causes of white matter lesions, which in some cases could disrupt the brain parenchyma and white matter tracts. We excluded patients who had prior history of neurological disorders or prior history of persistent neurological symptoms, as we already commented in the first version. We have further discussed this possible impact.
[Pages 14-15, lines 447-453] “Because of time restrictions related to the MRI acquisition protocol, it was not possible to collect T2 or T2-FLAIR MRI data to assess the presence of white matter hyperintensities. Pain in patients with EM and an adverse prognosis have been related to white matter hyperintensities in migraine [68,69]. Considering the relative high risk of detection of white matter hyperintensities on MRI in migraine and their negative impact, the presence of these lesions might have some influence on the results, although the associated pathophysiology and long-term effects are unclear [70].”
- Yalcin, A.; Ceylan, M.; Bayraktutan, O.F.; Akkurt, A. Episodic Migraine and White Matter Hyperintensities: Association of Pain Lateralization. Pain Med. 2018, 19, 2051–2057, doi:10.1093/pm/pnx312.
- Xie, H.; Zhang, Q.; Huo, K.; Liu, R.; Jian, Z.-J.; Bian, Y.-T.; Li, G.-L.; Zhu, D.; Zhang, L.-H.; Yang, J.; et al. Association of white matter hyperintensities with migraine features and prognosis. BMC Neurol 2018, 18, 93, doi:10.1186/s12883-018-1096-2.
- Porter, A.; Gladstone, J.P.; Dodick, D.W. Migraine and white matter hyperintensities. Curr Pain Headache Rep 2005, 9, 289–293, doi:10.1007/s11916-005-0039-y.
- Very few patients had aura. Why not exclude them?
R1.5. Thank you for the suggestion. By suggestion from Reviewer #2, we have corrected the results by the presence of aura, instead of excluding them. We have chosen this option because the loss of statistical power associated with a lower number of subjects could have an important impact on the results, perhaps higher than the proper effect of aura itself. Considering only the effects of aura, most AD and the RTPP results remained statistically significant, while the RTOP results (lower RTOP in episodic migraine compared to controls in 24 regions) lost the statistical significance. Moreover, following the suggestions from Reviewer #2, we also corrected the results including duration of migraine and medication overuse headache as covariates. We show below the aura results (second paragraph) and all the results with multiple covariates.
[Page 11, lines 298-322] “Taking into account the significant differences of presence of aura and medication overuse headache between patients with CM and EM, we additionally included both variables in the analysis with multiple covariates, together with the duration of migraine history.
Including only the presence of aura as a covariate, with respect to the results with no covariates, AD values were lower in CM than in EM in eight regions, the RTPP was higher in CM than in EM in the middle cerebellar peduncle, and no significant RTOP differences between EM and HC were obtained. No additional significant results were identified.
Regarding significant differences between CM and EM in the multivariate model, AD (middle cerebellar peduncle), MD (34 regions) and RD (39 regions) values were lower in CM. The opposite statistically significant trend, i.e., higher values in CM compared to EM, was obtained for RTPP (15 regions), RTOP (42 regions), and RTAP (38 regions). These results are shown in Tables S6-11 and Figures S3-5. Similar results, with the same significant statistical comparisons but a different number of regions with significant differences, were obtained including only medication overuse headache as a covariate. In addition to the previous results, including only medication overuse headache as a covariate, FA values were higher in CM compared to EM in the body (60 voxels, minimum adjusted p = 0.047) and splenium (136 voxels, minimum adjusted p = 0.040) of the corpus callosum, the left superior corona radiata (68 voxels, minimum adjusted p = 0.047), and the left tapetum (35 voxels, minimum adjusted p = 0.040).
Statistically significant differences employing the model with the three covariates were obtained between EM and HC. On the one hand, increased AD (eight regions) values were found in EM with respect to HC. On the other hand, reduced RTPP (five regions) values were identified in EM compared to HC. These results are shown in Tables S12-13 and Figure S4.
In the comparison between CM and HC, significantly higher FA (10 regions) values were found in CM, and significantly lower RD (14 regions) values were identified in CM. These results are shown in Tables S14-15 and Figure S5.”
We have also commented these new results with more detail in the Discussion.
[Page 14, lines 407-424] “Regarding previous DTI results in migraine, the reduction of FA in patients with migraine compared to controls is the most frequent result in diffusion MRI studies [2–5,38,46,49,61–65], but the opposite result has also been reported in pediatric patients by Messina et al. [6] and in the thalamus by Coppola et al. [66]. In this study, we obtained both higher and lower FA values in patients with CM compared to controls, associated with medication overuse headache and duration of migraine, respectively. Most regions related to both results were different, and the related clusters were not extremely close. A possible reason of these apparently contradictory results would be the coexistence of debilitated and enhanced structural networks in migraine, as it has been previously suggested [17], which may partially explain the apparently conflicting results in the literature. According to our results, these networks would be related to different pathophysiological mechanisms associated with medication overuse and longitudinal effects. In patients with EM and medication overuse headache, lower FA values have been reported compared to controls [5]. Additional significant differences were obtained between CM, EM and HC including duration of migraine and medication overuse headache, but no changes were specifically related to the presence of aura, in contrast to previous studies [67]. The lack of significant results associated with aura in our study may be caused by the relative low number of patients with aura. Future studies should specifically analyze differences between patients with and without medication overuse headache in patients with CM, and longitudinal studies should be performed to assess the longitudinal effects of CM in white matter.”
- Was any standard, structural/clinical assessment of scans performed? Ie. were there any radiological findings?
R1.6. Thank you for the question. All T1-weighted images were reviewed by a radiologist (MR, one of the coauthors). Subjects with brain abnormalities were not included in the initial sample. Hence, no radiological findings were observed in our sample.
[Page 3, lines 106-107] “No participants with brain abnormalities detected on T1-weighted MRI data by a radiologist were included in the sample.”
- The authors state that the sample did not contain patients with depression or anxiety (pg 12 ln 351) yet on pg 13 ln 366 it is stated that the presence of this was not assessed. It is not completely clear what is meant.
R1.7. The reviewer is right. We simply meant that there were no patients with depression or anxiety in our sample, and therefore its possible effects on the diffusion measures could not be studied. Previously, depression has been associated with smaller brain volume in patients with migraine, so it could be interesting to look for a possible relationship between MRI findings and depression and/or anxiety in migraine. In a previous study, migraine with depression has been suggested as a distinct clinical phenotype (reference at the end of this answer). In our case, as no patients in this situation were present, we avoided a possible bias caused by these factors. With respect to the sample, we excluded patients with major psychiatric disorders, in anamnesis or following the depression scores from the Hospital Anxiety and Depression Scale, as already mentioned in the first version of the manuscript. We have explained this point with more detail.
[Page 15, lines 464-471] “The presence of anxiety and/or depression, which are frequent in migraine, might influence brain connectivity, as pointed by previous studies. Indeed, smaller brain volume has been associated with depression in migraine, and migraine with depression may represent a different clinical phenotype with a specific long-term evolution [73]. In our sample, however, there were no patients with anxiety and/or depression. Although this fact prevented us from studying this possible relationship, the absence of patients with depression or anxiety also avoids the possible bias in the results that could have been caused by the inclusion of this distinct phenotype.”
- Gudmundsson, L.S.; Scher, A.I.; Sigurdsson, S.; Geerlings, M.I.; Vidal, J.-S.; Eiriksdottir, G.; Garcia, M.I.; Harris, T.B.; Kjartansson, O.; Aspelund, T.; et al. Migraine, depression, and brain volume: the AGES-Reykjavik Study. Neurology 2013, 80, 2138–2144, doi:10.1212/WNL.0b013e318295d69e.

Reviewer 2 Report
This study compared 3 groups, HC, EM and CM. However, EM and CM have significant difference in migraine disease duration, medication overuse and presence of aura. These factors should be evaluated by multiple variate analysis.
Migraine disease duration and each DTI parameter should be analyzed using scatter plot analysis.
Only middle cerebellar peduncle shower higher RTPP in MC than EM. The reason and significance of this finding should be discussed.
As author pointed out, there are many limitation in this study. Some of these limitation can be resolved analyzing depression status and headache diary of the participant in this study.
Author concluded these DIT finding is caused by migraine, without enough data and discussion.
Author Response
REVIEWER: 2
Thank you for all your suggestions and comments. Thanks to them, the manuscript has improved with a more detailed discussion, and we could identify new results that are important in our manuscript.
Reviewer #2:
- This study compared 3 groups, HC, EM and CM. However, EM and CM have significant difference in migraine disease duration, medication overuse and presence of aura. These factors should be evaluated by multiple variate analysis.
R2.1. Thank you for your valuable suggestion. We have now performed and included the analysis with the three covariates, obtaining interesting new results. As we obtained with the AD uncorrected results, lower MD and RD values were obtained in CM compared to EM in multiple regions (34 and 39 regions, respectively). Regarding the EAP-based measures, significantly higher RTOP, RTAP and RTPP values in CM with respect to EM in 42, 38, and 15 regions, respectively. Similar results, i.e., the same trends with significant differences but a different number of regions, were obtained when including only medication overuse headache as a covariate. In addition to the comparison between CM and EM, including the three covariates, increased AD values in EM compared to HC (similar results compared to the model including only the duration of migraine as a covariate), lower RTPP values in EM with respect to HC (the complementary results to the AD values), and higher FA values in CM than in HC (in contrast to the model including only time from onset of CM as a covariate) were obtained. Considering these new results, we have assessed the effect of these covariates with more detail in the Discussion, remarking the importance of medication overuse headache and duration of migraine in the assessment of white matter. Regarding the effect of the presence of aura, we did not observe new associated results, but it must be noted that this lack of significant results may be related to the low number of patients with aura in our sample.
[Page 11, lines 298-322] “Taking into account the significant differences of presence of aura and medication overuse headache between patients with CM and EM, we additionally included both variables in the analysis with multiple covariates, together with the duration of migraine history.
Including only the presence of aura as a covariate, with respect to the results with no covariates, AD values were lower in CM than in EM in eight regions, the RTPP was higher in CM than in EM in the middle cerebellar peduncle, and no significant RTOP differences between EM and HC were obtained. No additional significant results were identified.
Regarding significant differences between CM and EM in the multivariate model, AD (middle cerebellar peduncle), MD (34 regions) and RD (39 regions) values were lower in CM. The opposite statistically significant trend, i.e., higher values in CM compared to EM, was obtained for RTPP (15 regions), RTOP (42 regions), and RTAP (38 regions). These results are shown in Tables S6-11 and Figures S3-5. Similar results, with the same significant statistical comparisons but a different number of regions with significant differences, were obtained including only medication overuse headache as a covariate. In addition to the previous results, including only medication overuse headache as a covariate, FA values were higher in CM compared to EM in the body (60 voxels, minimum adjusted p = 0.047) and splenium (136 voxels, minimum adjusted p = 0.040) of the corpus callosum, the left superior corona radiata (68 voxels, minimum adjusted p = 0.047), and the left tapetum (35 voxels, minimum adjusted p = 0.040).
Statistically significant differences employing the model with the three covariates were obtained between EM and HC. On the one hand, increased AD (eight regions) values were found in EM with respect to HC. On the other hand, reduced RTPP (five regions) values were identified in EM compared to HC. These results are shown in Tables S12-13 and Figure S4.
In the comparison between CM and HC, significantly higher FA (10 regions) values were found in CM, and significantly lower RD (14 regions) values were identified in CM. These results are shown in Tables S14-15 and Figure S5.”
[Page 14, lines 407-424] “Regarding previous DTI results in migraine, the reduction of FA in patients with migraine compared to controls is the most frequent result in diffusion MRI studies [2–5,38,46,49,61–65], but the opposite results has also been reported in pediatric patients by Messina et al. [6] and in the thalamus by Coppola et al. [66]. In this study, we obtained both higher and lower FA values in patients with CM compared to controls, associated with medication overuse headache and duration of migraine, respectively. Most regions related to both results were different, and the related clusters were not extremely close. A possible reason of these apparently contradictory results would be the coexistence of debilitated and enhanced structural networks in migraine, as it has been previously suggested [17], which may partially explain the conflicting results in the literature. According to our results, these networks would be related to different pathophysiological mechanisms associated with medication overuse and longitudinal effects. In patients with EM and medication overuse headache, lower FA values have been reported compared to controls [5]. Additional significant differences were obtained between CM, EM and HC including duration of migraine and medication overuse headache, but no changes were specifically related to the presence of aura, in contrast to previous studies [67]. The lack of significant results associated with aura in our study may be caused by the relative low number of patients with aura. Future studies should specifically analyze differences between patients with and without medication overuse headache in patients with CM, and longitudinal studies should be performed to assess the longitudinal effects of CM in white matter.”
- Migraine disease duration and each DTI parameter should be analyzed using scatter plot analysis.
R2.2. We have added the association between diffusion (DTI and AMURA) parameters and duration of migraine (years). In the case of patients with chronic migraine, we have additionally assessed the time from onset of chronic migraine (months). We found statistically significant positive correlation between FA and time from onset of chronic migraine in the bilateral external capsule, and negative correlation between RD and time from onset of chronic migraine in the bilateral external capsule. These significant results have been previously published in a previous paper (reference number 7). Regarding the total duration of migraine, other DTI parameters (MD and AD), and the AMURA parameters, no significant correlations were identified.
[Page 5, lines 206-215] “Moreover, the association between duration of migraine (total duration or time from onset of CM) and DTI and AMURA measures was assessed. To analyze trends within each type of migraine and following the previous study with the same sample [7], we acquired the correlation values in patients with EM and CM independently. The inverse warp fields of the FA images to the MNI space transformation from the TBSS processing steps were obtained and used to obtain individual label maps based on the John Hopkins University ICBM-DTI-81 White-Matter atlas. The mean value of each parameter in the diverse regions of the atlas and the Spearman’s rank correlation value were employed in the correlation analysis. The results were corrected for multiple comparisons with the Benjamini-Hochberg false discovery rate method [37]. A value of p < 0.05, adjusted for multiple comparisons, was considered statistically significant.”
[Pages 11-12, lines 322-327, 333-334] “Statistically significant positive correlation was identified between time from onset of CM and mean FA in the left (ρ = 0.439, unadjusted p < 0.001) and right (ρ = 0.420, unadjusted p = 0.001) external capsule after the correction for multiple comparisons. Statistically significant negative correlation was found between time from onset of CM and mean RD in the left (ρ = -0.439, unadjusted p < 0.001) and right (ρ = -0.427, unadjusted p = 0.001) external capsule. The significant correlation results are shown in Figure 6 (extracted from [7]).
No additional statistically significant correlations were found for mean AD or MD, the three EAP-based measures, or the total duration of the migraine in patients with EM or CM.”
- Planchuelo-Gómez, Á.; García-Azorín, D.; Guerrero, Á.L.; Aja-Fernández, S.; Rodríguez, M.; de Luis-García, R. White matter changes in chronic and episodic migraine: a diffusion tensor imaging study. J Headache Pain 2020, 21, 1, doi:10.1186/s10194-019-1071-3.
37. Benjamini, Y.; Hochberg, Y. Controlling the false discovery rate: A practical and powerful approach to multiple testing. J. R. Stat. Soc. Ser. B 1995, 57, 289–300, doi:10.2307/2346101.
- Only middle cerebellar peduncle shower higher RTPP in MC than EM. The reason and significance of this finding should be discussed.
R2.3. Thank you for the suggestion. It must be noted that RTPP results are expected to be related to the AD results, as their interpretation is similar, despite being different measures, because they measure diffusion components perpendicular to the main direction. RTPP results present higher p-values and follow opposite trends compared to AD results (in Appendix A the reason of the opposite trends can be clearly observed). The RTPP values were obtained in suboptimal conditions for the AMURA toolbox (single-shell and low b-value acquisition), which possibly explains the lower number of regions with significant differences compared to the AD results. Furthermore, regarding the biological reason, we have associated our results with alterations of the cerebellum and pons in migraine, considering that the middle cerebellar peduncle connects these two regions.
[Page 13, lines 361-379] “The significant differences between patients with CM and EM obtained with the RTPP were a subsample of the results obtained with the AD. It must be considered that the diffusion MRI acquisition protocol was suboptimal for the use of the AMURA tool because of the low b-value and single-shell scheme, and, possibly for that reason, we were unable to identify the same or a similar number of significant differences between the migraine groups. The three measures obtained with the AMURA tool are designed to measure effects related to b-values over 2000 s/mm2 (our b-value was 1000 s/mm2), and RTPP is particularly sensitive to the b-value. Therefore, the RTPP result reflects the potential of AMURA to identify white matter structural changes in migraine using an acquisition protocol more appropriate for the tool.
RTPP differences between patients with CM and EM were found in the middle cerebellar peduncle. The middle cerebellar peduncle connects the cerebellum to the pons. Smaller cerebellar volume has been identified in patients with CM compared to HC [42]. Moreover, it has been suggested that the cerebellum suffers a neuropathological change in migraine related to spreading depression [43]. The dorsolateral pons has been shown to be activated during migraine and potentially involved in other mechanisms such as transmission of nociceptive signals to the hypothalamus, amygdala, and basal forebrain [44]. Another study by Chong et al. has reported significant deformation of the pons in patients with migraine [45]. Our results may suggest that the connectivity between the cerebellum and the pons is altered in CM compared to EM, possibly in association with structural changes of these regions linked to the migraine experience.”
- Bilgiç, B.; Kocaman, G.; Arslan, A.B.; Noyan, H.; Sherifov, R.; Alkan, A.; Asil, T.; Parman, Y.; Baykan, B. Volumetric differences suggest involvement of cerebellum and brainstem in chronic migraine. Cephalalgia 2016, 36, 301–308, doi:10.1177/0333102415588328.
- Vincent, M.; Hadjikhani, N. The cerebellum and migraine. Headache 2007, 47, 820–833, doi:10.1111/j.1526-4610.2006.00715.x.
- Borsook, D.; Burstein, R. The enigma of the dorsolateral pons as a migraine generator. Cephalalgia 2012, 32, 803–812, doi:10.1177/0333102412453952.
- Chong, C.D.; Plasencia, J.D.; Frakes, D.H.; Schwedt, T.J. Structural alterations of the brainstem in migraine. NeuroImage. Clin. 2016, 13, 223–227, doi:10.1016/j.nicl.2016.10.023.
- As author pointed out, there are many limitation in this study. Some of these limitation can be resolved analyzing depression status and headache diary of the participant in this study.
R2.4. With respect to the depression status, we did not include these patients to avoid a possible bias. It has been suggested that migraine with depression may represent a distinct phenotype in relation to MRI findings related to smaller brain volume in these patients (the reference is shown at the end of this answer). Regarding other limitations, it is true that the diagnosis of tension-type headache or the control of the time to the migraine attack after the MRI acquisition might have been resolved with more detailed information in the headache diary, but we do not have the corresponding retrospective data. We have explained with more detail the depression limitation.
[Page 15, lines 464-471] “The presence of anxiety and/or depression, which are frequent in migraine, might influence brain connectivity, as pointed by previous studies. Indeed, smaller brain volume has been associated with depression in migraine, and migraine with depression may represent a different clinical phenotype with a specific long-term evolution [73]. In our sample, however, there were no patients with anxiety and/or depression. Although this fact prevented us from studying this possible relationship, the absence of patients with depression or anxiety also avoids the possible bias in the results that could have been caused by the inclusion of this distinct phenotype.”
- Gudmundsson, L.S.; Scher, A.I.; Sigurdsson, S.; Geerlings, M.I.; Vidal, J.-S.; Eiriksdottir, G.; Garcia, M.I.; Harris, T.B.; Kjartansson, O.; Aspelund, T.; et al. Migraine, depression, and brain volume: the AGES-Reykjavik Study. Neurology 2013, 80, 2138–2144, doi:10.1212/WNL.0b013e318295d69e.
- Author concluded these DIT finding is caused by migraine, without enough data and discussion.
R2.5. The reviewer is right. We have added a paragraph to discuss the specificity of changes related to migraine with respect to other headache disorders, including the reference to a recent study that included patients with diverse headache disorders. In that study, no significant differences were identified between migraine and tension-type headache employing TBSS to assess white matter changes. Moreover, it should be clarified whether these changes are cause or consequence of the migraine. Considering this last point, we have changed “caused by” into “related to” in the conclusion.
[Page 14, lines 433-446] “Regarding the alterations found in this study, it should be elucidated whether the identified changes are migraine-specific. In a previous study including 277 headache free subjects and 246 patients with headache, including 69 patients with migraine and 76 with tension-type headache, Kattem Husøy et al. found no significant TBSS differences between migraine and tension-type headache [49]. In the same study, the authors identified significant higher AD values in patients with migraine and tension-type headache compared to headache free subjects, with higher number of voxels with significant differences in migraine. Furthermore, patients with any headache and new onset headache presented widespread higher AD, MD and RD values compared to controls in the aforementioned study. These results suggest alterations in patients with headache compared to controls, but with no clear migraine biomarkers in contrast to other headache disorders. The specific migraine microstructural brain changes in comparison with other headache and pain disorders should be analyzed in future studies, in order to uncover the particular pathophysiological characteristics of migraine. Another aspect that needs to be studied is whether the identified changes are cause or consequence of migraine.”
49. Kattem Husøy, A.; Eikenes, L.; Håberg, A.K.; Hagen, K.; Stovner, L.J. Diffusion tensor imaging in middle-aged headache sufferers in the general population: a cross-sectional population-based imaging study in the Nord-Trøndelag health study (HUNT-MRI). J Headache Pain 2019, 20, 78, doi:10.1186/s10194-019-1028-6.

Round 2
Reviewer 2 Report
The manuscript was significantly improved.